# Quantifying Urban Linguistic Diversity Related to Rainfall and Flood across China with Social Media Data

Jiale Qian [1,2], Yunyan Du [1,2,*], Fuyuan Liang [1], Jiawei Yi [1,2], Nan Wang [1,2], Wenna Tu [1,2], Sheng Huang [1,2], Tao Pei [1,2] and Ting Ma [1,2]

1   State Key Laboratory of Resources and Environmental Information System, Institute of Geographical Sciences and Natural Resources Research, Chinese Academy of Sciences, Beijing 100101, China; qianjl@lreis.ac.cn (J.Q.)
2   University of Chinese Academy of Sciences, Beijing 100049, China
*   Correspondence: duyy@lreis.ac.cn

**Abstract:** Understanding the public's diverse linguistic expressions about rainfall and flood provides a basis for flood disaster studies and enhances linguistic and cultural awareness. However, existing research tends to overlook linguistic complexity, potentially leading to bias. In this study, we introduce a novel algorithm capturing rainfall and flood-related expressions, considering the relationship between precipitation observations and linguistics expressions. Analyzing 210 million social media microblogs from 2017, we identified 594 keywords, 20 times more than usual manually created bag-of-words. Utilizing Large Language Model, we categorized these keywords into rainfall, flood, and other related terms. Semantic features of these keywords were analyzed from the viewpoint of popularity, credibility, time delay, and part-of-speech, finding rainfall-related terms most common-used, flood-related keywords often more time delayed than precipitation, and notable differences in part-of-speech across categories. We also assessed spatial characteristics from keyword and city-centric perspectives, revealing that 49.5% of the keywords have significant spatial correlation with differing median centers, reflecting regional variations. Large and disaster-impacted cities show the richest expression diversity for rainfall and flood-related terms.

**Keywords:** linguistic diversity; flood; rainfall; social media; Chinese dialects; bias

## 1. Introduction

Language is a fundamental and intricate facet of human communication, playing a crucial role in the exchange of thoughts, emotions, and information [1–4]. Linguistics is the scientific study of language, aiming to objectively examine its structure, use, and societal role, thus addressing a broad range of questions about human language [5–8]. Linguistic diversity, reflecting intricate variations in language structures and patterns, is a dynamic area of study within linguistics [9–12]. For example, linguistic preferences for carbonated beverages vary significantly across the U.S.: it is commonly referred to as a "soft drink" in general parlance, termed "pop" in the Midwest, dubbed "soda" in the Northeast, and colloquially called "Coke" in the South—a term applied irrespective of the brand [13,14]. Linguistic diversity paints a fascinating mosaic, reflecting the unique histories, geographies, and sociocultural contexts of various populations [15,16]. Studying linguistic diversity has profound implications for our understanding of human cognition and the broader societal landscape. Linguistic diversity highlights the dynamic interplay between language and identity, fostering cultural preservation, intercultural dialogue, and global connections [17,18].

The increased emphasis on language and culture preservation has expanded research on linguistic diversity, especially within diverse linguistic and cultural terrain in China [19,20]. Previous research predominantly used statistics and surveys, employing oral interviews, questionnaires, and field investigations for data collection and

analysis [21–23]. Although these methods are valuable, they also have inherent limitations. Due to resource and time constraints, these studies are often smaller in scale, focusing on specific regions or communities. Additionally, such studies may have a limited scope, possibly prioritizing certain languages or dialects over others. This narrow focus might result in research outcomes that do not fully capture the linguistic diversity in Chinese urban contexts. Moreover, traditional research methods may be prone to biases. For instance, researchers might favor languages or dialects they are familiar with or find interesting, overlooking others.

The rise of big data from social media has significantly evolved the methods and focus of linguistic diversity research [24–30]. Blodgett, Green [31] used Twitter geolocation data to explore dialectal variations in African-American English. Using a distant supervision model, they identified texts linked to African-Americans and verified their alignment with known African-American English patterns. Sadat, Kazemi [32] sought to bridge the gap between Modern Standard Arabic (MSA) and Arabic Dialects (AD) with a translation framework for social media texts. The study used linguistic tools, such as a bilingual AD-MSA lexicon and grammatical mapping rules, to convert Tunisian dialect sentences to MSA. Although these studies provide valuable insights, their limitations should be acknowledged. Primarily, current research has not deeply delved into environmental and public crises domains, such as linguistic diversity related to rainfall and floods, underscoring the need for thorough exploration. Secondly, many research methodologies focus on internal linguistic analysis, neglecting the integration of data from various sources.

The public crisis domain stands as a pivotal area of investigation in linguistic research [33–36]. Many researchers utilize social media data, as a medium of language, to detect crises [37–39], analyze resilience [40–44], gauge situational awareness [29,45,46], and define crisis thresholds [47,48]. A foundational step in these studies involves extracting and identifying microblogs related to rainfall and flood, typically using two approaches: bag-of-words filtering and machine learning-based annotation prediction. For example, Wang, Loo [49] utilized five key terms such as "雨" (rain in Chinese) and "洪水" (flood in Chinese), derived from media reports on urban flooding and trial data runs, to extract flood-related microblogs. Said, Ahmad [50] employed an Italian version of BERT, training the model with labeled samples for text classification. Despite these advancements in integrating linguistic perspectives into disaster analysis and the effective extraction of social media data related to specific regions and events, there are several limitations. First, the keywords for rainfall and flood are often used interchangeably as linguistic representations of flood disasters, but terms such as rainfall only stand for weather and meteorological conditions. Only terms such as flood and inundation are the direct indicators of the affected areas. Second, the selection of keywords or labeled sample texts is manually performed, leading to incomplete selection and potential extraction bias due to the lack of automation, ultimately causing regional biases and disparities. Lastly, while some keywords have global usage, some others are region-specific, limited to certain dialectical areas. Although such globally recognized keywords are frequently used [51], treating it as a proxy for the entire rainfall or flood term may lead to biased results. In other words, the approach does not adequately account for linguistic diversity.

Rainstorms and floods serve as an ideal context for linguistic diversity studies due to their significant financial impacts, especially in countries such as China [52,53]. As the effects of climate change intensify, areas previously untouched by floods now face increased threats, with expectations of amplified risks from more frequent extreme precipitation events [54–58]. Using Typhoon Doksuri as a case study, this event brought numerous rainstorms and floods. The typhoon affected over a third of China's provinces, ranging from southern provinces such as Guangdong and Fujian to northern ones such as Beijing and Heilongjiang. Spanning over 9.6 million km$^2$, China's vastness encompasses over 80 dialects [59]. Therefore, neglecting linguistic diversity in research or disaster management could lead to regional biases and cause inequalities.

Our study sets out to provide a comprehensive quantification of rainfall- and flood-related keywords. This research offers three major contributions to the discipline. First, we introduce a novel quantitative algorithm that captures expressions related to rainfall and floods by considering the nexus between precipitation observations and linguistic phrases. Implementing this approach on 210 million social media entries from 2017, we identified 594 keywords associated with rainfall and flooding. This count is 20-fold higher than the manually created bag-of-words, thereby significantly enriching the keyword library. Second, we delve into a thorough analysis of these keywords' semantic attributes, exploring aspects such as popularity, credibility, and time delay. Third, we dissect spatial characteristics from dual viewpoints: the keyword perspective (considering each keyword's regional applicability and its global or local nature) and the city-centric perspective (assessing keyword diversity within individual cities). Furthermore, we discuss the driving forces behind these distributions, enabling us to offer insights that can guide local authorities and residents in formulating nuanced strategies and fostering a sustainable habitat. Our findings serve as a robust foundation for future studies leveraging social media data to gauge public perception of rainfall and flooding, offering enhanced accuracy and completeness especially in linguistically diverse settings such as China.

## 2. Data

From the Sina Weibo platform—a leading social media outlet in China with over 165 million daily users—we gathered more than 210.8 million geotagged microblogs from 2017 (Table 1). The dataset encompasses geotagged content, message IDs, user-reported locations, timestamps, user IDs, follower counts, and retweet metrics. All microblogs underwent anonymization before dataset compilation to ensure privacy protection. We also procured the 2017 Version06 GPM IMERG 30 min precipitation dataset [60] with a spatial resolution of $0.1° \times 0.1°$. The dataset, renowned for its accuracy, has featured prominently in prior research, such as [61,62]. Prior studies have confirmed that rainfall in 2017 was exceptionally varied, spanning a range of intensities and types—from light showers to severe floods across different regions [36]. This suggests that our choice to focus on data from 2017 provides a representative foundation for analyzing public perceptions and linguistic diversity of rainfall and flooding events.

**Table 1.** Data description.

| Data | Website | Resolution | Data Time |
|---|---|---|---|
| Weibo | https://open.weibo.com (accessed on 22 May 2021) | Points/s | 2017 |
| GPM | https://gpm.nasa.gov/data/imerg (accessed on 11 February 2022) | 0.1°/30 min | 2017 |

## 3. Method

The methodological framework of the study is organized into four main components, as illustrated in Figure 1. Initially, the focus was on preprocessing rainfall and flood-related microblogs, which included filtering out characters, advertisements, and applying spatial filtering. Following this, precipitation data and microblog content were transformed into daily count vectors. In the third phase, the similarity between each pair of vectors was evaluated to identify all precipitation-related expressions, which were then classified into three categories: rainfall, flood, and other related terms, utilizing a Large Language Model. The final component of our framework involved a spatial analysis of rainfall and flood-related keywords, which encompassed assessing spatial correlations and identifying central locations.

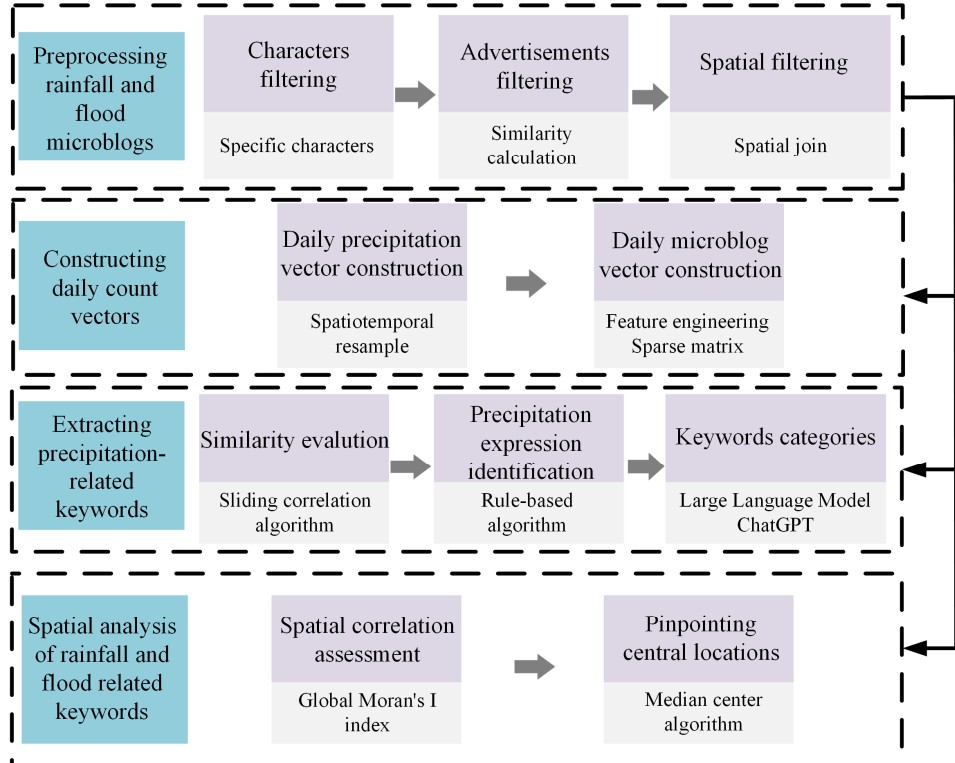

**Figure 1.** Color-coded methodological framework: blue for section topics, purple for processes, and grey for detailed methods.

### 3.1. Preprocessing Rainfall and Flood Microblogs

We began by isolating microblogs from the 210 million posted across China in 2017 that contained characters associated with rainfall or flooding, specifically "洪/涝/雨/水/海/汛" (equivalent to rainfall and flood in English). We utilized the jieba toolkit—a prevalent Chinese word segmentation technology—to transform the Chinese text into word sequences, also labeling the part-of-speech for each term.

Our filtering methodology encompassed two primary stages. Initially, we targeted and eliminated users primarily posting advertisements. Such users often exhibit a high posting frequency with substantial content consistency. We identified them by seeking any two microblogs with over 80% similarity in their postings. Detected users were deemed as advertising accounts, and their associated microblogs were excluded from our dataset.

Subsequently, we refined our dataset to exclude microblogs originating outside of China. Utilizing the spatial join functionality in ArcGIS, we matched each microblog's longitude and latitude which were reported by user to a China basemap encompassing 369 cities, retaining only the corresponding entries.

Ultimately, we obtained a refined dataset comprising 35,521,272 microblogs, ready for further keyword analysis. The distribution of microblog counts across cities is illustrated in Figure S1. The number of microblogs from various cities all exceed 100, demonstrating regional representativeness.

### 3.2. Constructing Daily Count Vectors for Precipitation and Microblogs

Prior to assessing similarity, our initial step involved constructing daily count vectors for both precipitation and the microblogs linked to each keyword ($MB_k$). For the precipitation data, we resampled the half-hourly GPM data to a daily metric, subsequently computing the average daily rainfall across each city. This yielded a 365-element vector denoting precipitation for every city. In the realm of microblog data, we began by isolating every $MB_k$. We had over 500,000 candidate keywords after filtering. To expedite computation, we employed feature engineering as an alternative to direct text matching. Using the

outcomes from the jieba word segmentation, we formulated a feature engineering model. In this model, individual rows signify distinct microblogs, columns stand for keywords, and cells highlight the occurrence count of a particular keyword within a given microblog. To optimize storage efficiency, we saved this matrix using a sparse matrix configuration. For every keyword, we extracted the microblog indices containing that keyword from the sparse matrix, thus pinpointing $MB_k$. Conclusively, we consolidated the timestamps associated with $MB_k$ to a daily scale, producing a daily $MB_k$ tally. This process culminated in generating a 365-element vector showcasing microblog counts tailored for each city and keyword.

### 3.3. Extracting Precipitation-Related Keywords

Theoretically, when precipitation leads to flooding, subsequent relief and aid activities ensue post the precipitation peak. To holistically identify linguistic expressions tied to rainfall and flooding, we devised an algorithm dedicated to filtering precipitation-related keywords. Initially, we employed the sliding correlation algorithm [63], a method prevalent in time series analysis and signal processing, to discern the correlation between the precipitation and $WB_K$ time series. This method identifies time lags or delays between sequences by pinpointing the highest similarity between two-time sequences and establishing their time difference. The essence of sliding correlation is determining the correlation coefficient of two sequences; one remains static while the other slides, enabling correlation coefficient calculation at diverse time junctures. For our research, our sliding window encompassed 11 distinct categories, spanning from $-5$ to 5 days. Our comprehensive empirical observations indicated that keywords tied to precipitation or with a delayed correlation exhibited a $p$-value curve resembling either an L or U shape. These started as non-significant but soon became significant, persisting for a minimum of three consecutive units (days). Drawing from these insights, we formulated a rule-based algorithm to pinpoint keywords intrinsically linked to actual precipitation. The precise steps for screening encompassed the following: We analyzed the slide points in ascending order. If the $p$-value at a particular slide point exceeded 0.05 and the subsequent three $p$-values were all below 0.05, the keyword was deemed precipitation-related for the city. To characterize such keywords, we introduced two metrics: the Maximum Coefficient of Correlations (MCC) and the Optimal Latency (OL). We determined the MCC by finding the highest correlation coefficient among points with a $p$-value below 0.05, where the x-coordinate of this point indicated the OL. Upon identifying all relevant keywords, we utilized Large Language Model ChatGPT to categorize these keywords into rainfall, flood, and other related terms.

### 3.4. Spatial Analysis of Rainfall and Flood Related Keywords

After filtering precipitation-related keywords, we identified the significant keywords associated with each city. In other words, we could map the city distribution for each of these significant keywords. To conduct a rigorous analysis of the spatial characteristics of cities that are significantly correlated with each keyword, we employed two methodologies: the Global Moran's I index [64] for spatial correlation assessment and the median center algorithm [65] for pinpointing central locations. The Global Moran's I index requires two critical inputs: attribute values and a spatial weight matrix. In our setup, cities significantly related to a particular keyword were assigned a value of "1", while all other cities were marked "0". We derived our spatial weight matrix from the Queen's contiguity matrix [66], which accounts for both shared vertices and edge connections. For the computational aspects, we leveraged the spatial exploratory data analysis libraries esda and libpysal [67]. We used the median center algorithm to determine the central location of corresponding cities which had a value equal to "1" for each significant keyword. This algorithm calculates the median position across all cities significantly related to a particular keyword, providing a robust measure for understanding distribution patterns and identifying optimal locations within geographical regions.

## 4. Results

### 4.1. Rainfall and Flood Related Keyword Library

In 2017, Changsha City faced multiple intense precipitation episodes that led to rainfall and significant flooding. Figure 2 depicts the yearlong variation of both precipitation levels and the frequency of microblogs featuring eight typical keywords (MB$_k$), using Changsha as a reference point. The analysis reveals a weak correlation between the frequency of MB$_k$ such as "Water wave" (水波) or "Aquarius" (水瓶座) and precipitation patterns, suggesting these terms are not closely tied to rainfall events. Conversely, MB$_k$ such as "Rain" (雨) or "Downpour" (暴雨) align closely with precipitation trends, with concurrent spikes in both rainfall and related microblog mentions. Analysis from Table 2 indicates that precipitation-unrelated keywords fall into two categories: those with no significant correlation, such as "Water wave", and those with a notable negative correlation, such as "Aquarius". Precipitation-related keywords predominantly showcase one characteristic: a strong positive correlation, as seen with terms such as "Rain", "Downpour", "Flood" (洪水), and "Inundation" (淹). However, keywords such as "Rebuild" (重建) and "Supplies" (物资), often associated with flood recovery efforts, do not demonstrate a notable correlation with precipitation data.

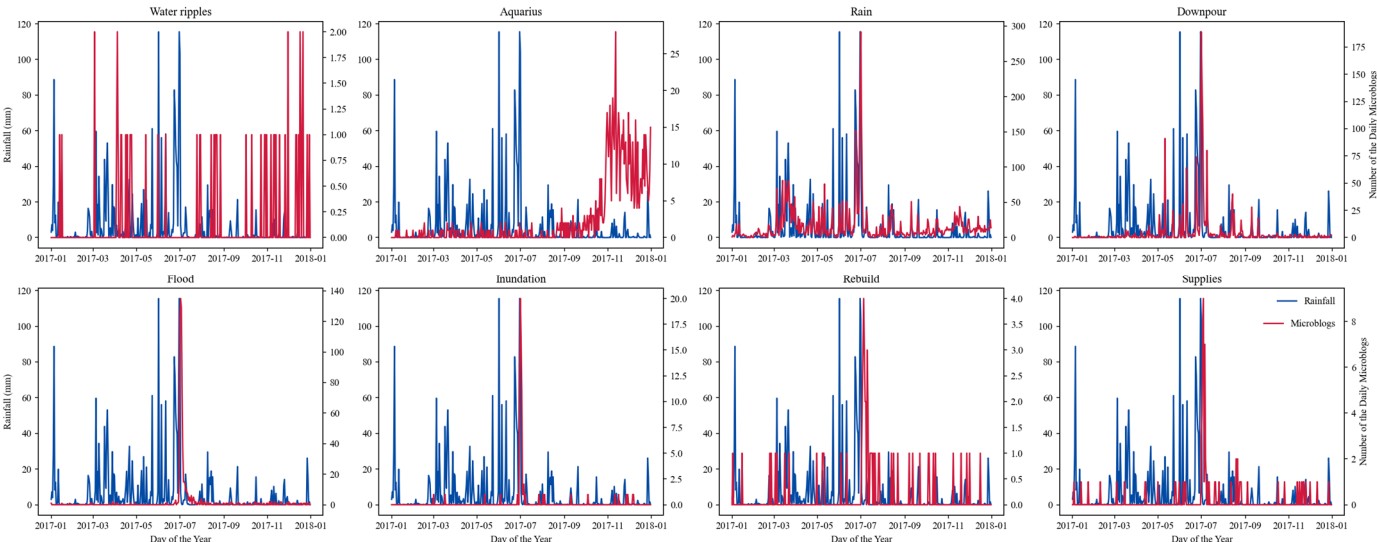

**Figure 2.** The annual variation curves of precipitation and the number of MB$_k$ at a daily time scale in Changsha city.

**Table 2.** Correlation analysis between the precipitation and the typical MB$_k$ in Changsha city.

| Keyword | Number of Microblogs | Pearson r |
|---|---|---|
| Water ripples | 52 | −0.06 |
| Aquarius | 896 | −0.14 ** |
| Rain | 6014 | 0.68 *** |
| Downpour | 1705 | 0.47 *** |
| Flood | 726 | 0.18 *** |
| Inundation | 81 | 0.36 *** |
| Rebuild | 59 | −0.03 |
| Supplies | 75 | 0.01 |

Notes: Asterisks (*) denote the levels of statistical significance of *p*-values, including ** $p < 0.01$, *** $p < 0.001$.

The sliding correlation coefficients between precipitation and the frequency of typical MB$_K$ in Changsha city are illustrated in Figure 3. Notably, *p*-value curves for keywords associated with rainfall and flooding adopt an L-shaped trajectory. Our precipitation-related keyword filtering algorithm outputs, as shown in Table 3, focus on two primary metrics:

Maximum Coefficient of Correlations (MCC) and Optimal Latency (OL). Words explicitly depicting precipitation, namely "Rain" and "Downpour", manifest a robust correlation with precipitation values of 0.68 and 0.47, respectively, without any noticeable latency. Keywords such as "Flood" and "Inundation", which directly refer to disasters, show significant correlations of 0.45 and 0.47, respectively, but with latencies of 1 and 2 days. Keywords indicating post-disaster scenarios, such as "Rebuild" and "Supplies", have their peak correlation at a 5-day latency, registering coefficients of 0.28 and 0.40, respectively.

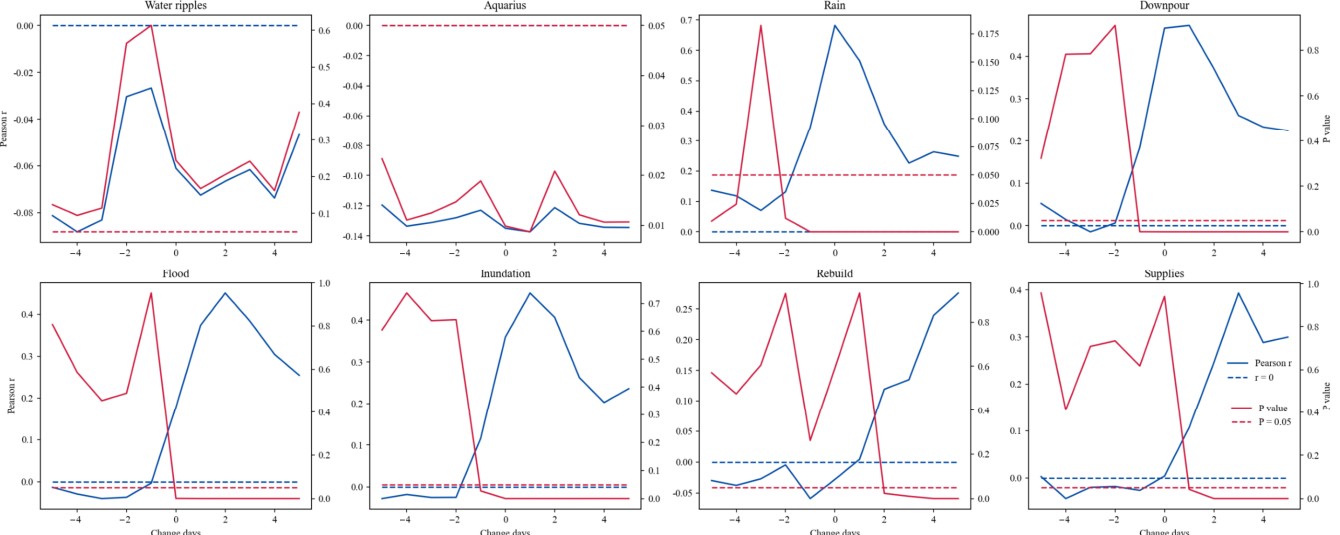

**Figure 3.** The sliding correlation coefficients between precipitation and the number of typical $MB_k$ in Changsha city.

**Table 3.** Algorithm results of keywords related to precipitation filtering.

| Keyword | Number of Microblogs | True or False | MCC | OL (Days) |
|---|---|---|---|---|
| Water ripples | 52 | False | | |
| Aquarius | 896 | False | | |
| Rain | 6014 | True | 0.68 | 0 |
| Downpour | 1705 | True | 0.47 | 1 |
| Flood | 726 | True | 0.45 | 2 |
| Inundation | 81 | True | 0.47 | 1 |
| Rebuild | 59 | True | 0.28 | 5 |
| Supplies | 75 | True | 0.40 | 3 |

Notes: In the True or False column, keywords retained after filtering through the rule-based algorithm are labeled as True; otherwise, they are labeled as False.

We subsequently identify all precipitation-related keywords in each city. Drawing from Large Language Model, we classify these keywords into three thematic categories. The first category, denoted as $KW_{rainfall}$, encompasses keywords directly referencing rainfall. The second category, labeled $KW_{flood}$, includes keywords specifically describing flooding disasters, their resulting disruptions to daily activities, property damages, and casualties. Finally, $KW_{other}$ represents the third category and comprises other keywords tied to rainfall and flooding. Figure 4 shows the word cloud distribution across keyword types. Our study finally obtains a total of 594 keywords, with $KW_{other}$ accounting for the highest number of keywords at 281, while $KW_{rainfall}$ and $KW_{flood}$ had similar numbers at 157 and 156, respectively. The number of these keywords is 20 times higher than the number of commonly used keywords in most studies, which greatly expands the keywords library.

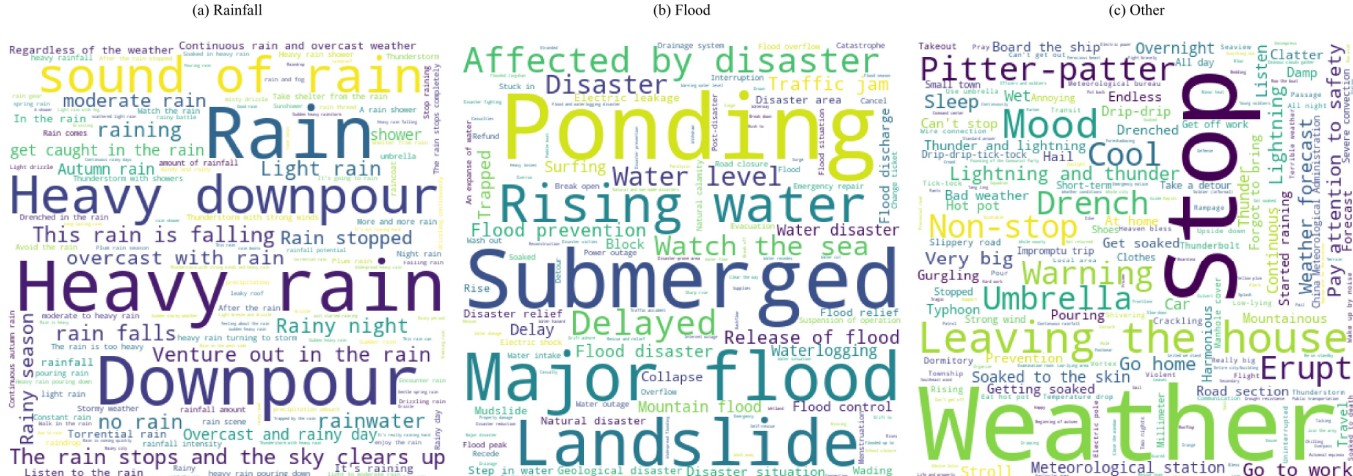

**Figure 4.** Word cloud distribution across keyword types with font size reflecting frequency.

### 4.2. Semantic Feature Variations of Keywords

A specific keyword can show a strong correlation with precipitation across different cities. We initially consolidated results from all cities with significant keyword correlations, formulating three metrics to signify popularity (number of cities), credibility (average MCC), and time delay (average OL). Figure 5 illustrates the distributional traits of these metrics for three primary thematic keywords, as well as the significant dissimilarities between the themes.

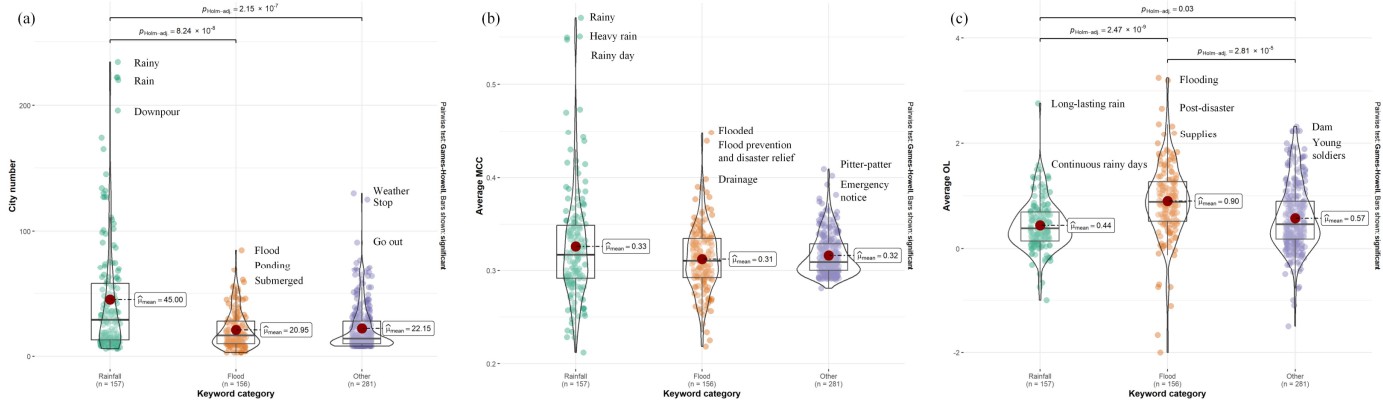

**Figure 5.** The distributional traits of (**a**) city numbers, (**b**) average MCC, and (**c**) average OL metrics across keyword types.

KW$_{rainfall}$ had the highest mean city numbers at 45, considerably higher than the other two categories, as depicted in Figure 5a. Among these, "Rainy" (下雨), "Rain" (雨), and "Downpour" (暴雨) had city numbers exceeding 200, indicating that they were the most widely utilized terms in different regions when referring to rainfall. KW$_{flood}$ had the lowest mean city numbers at 20.95; nevertheless, there was no significant variation in the average compared to KW$_{other}$. The keywords "Flood" (洪水), "Ponding" (积水), and "Submerged" (被淹) were the most commonly used terms in different regions when referring to floods. Furthermore, other terms such as "Weather" (天气), "Stop" (停), and "Go out" (出门) were frequently used during rainfall and floods.

Figure 5b shows that the average MCC of KW$_{rainfall}$, KW$_{flood}$, and KW$_{other}$ are all high, at 0.33, 0.31, and 0.32, respectively, with no significant differences between them. In KW$_{rainfall}$, "Rainy", "Heavy rain" (大雨), and "Rainy day" (下雨天) have the highest correlation coefficients, all exceeding 0.5, indicating the highest degree of credibility. In KW$_{flood}$,

"Flooded" (淹过), "Flood prevention and disaster relief" (防洪抗灾), and "Drainage" (排涝) have the highest credibility.

KW$_{flood}$ exhibits the highest average OL, at 0.9 days (Figure 5c), and is significantly greater than the other two categories. This indicates that the public perception of flood disasters tends to be delayed by approximately one day compared to the actual peak rainfall, mainly because the occurrence of floods is often related not only to the instantaneous intensity of precipitation but also to the cumulative rainfall. When the cumulative rainfall exceeds a certain threshold, the river water level rises, exceeds the warning level, and leads to dike breaches and dam failures, causing urban disasters. In KW$_{flood}$, "Flooding", "Post-disaster" (灾后), and "Supplies" (物资) are the terms with the longest delay days, all exceeding 2 days. In contrast, KW$_{rainfall}$ has the smallest average OL, at 0.44 days, indicating that the public perception of precipitation weather is almost in real-time. However, there are also some terms with a significant delay such as "Long-lasting rain" (久雨) and "Continuous rainy days" (连阴雨), which are mostly related to long-duration, low-intensity rainfall. In KW$_{flood}$, "Dam" (大坝) and "Young soldiers" (子弟兵) are also terms that are perceived relatively late.

Table 4 presents the results of part-of-speech (POS) analysis across different keyword types. Among these, the entropy for KW$_{rainfall}$ is the lowest at 1.48, indicating a low diversity in the types of POS. Specifically, the data reveal a predominance of nouns, accounting for 69% of the total words used for describing rainfall phenomena. This significantly outweighs the usage of other POS categories and suggests that terms such as "Heavy Rain" and "Downpour" are frequently used to convey rainfall events (Figure S2). In contrast, KW$_{flood}$ has a higher entropy value of 2.19, signifying a more complex and diverse distribution of POS types. Verbs constitute the largest portion at 45%, followed by nouns at 31%. This suggests that KW$_{flood}$ often employs action-oriented and process-focused terms such as "Submerge" (淹), "Interrupt" (中断), and "Paralyze" (瘫痪) to represent flooding events. KW$_{other}$ has the highest entropy value of 2.95, indicating a relatively even distribution of POS types without a clearly dominant category. Although nouns do make up 37%, verbs follow closely behind at 26%. This variation suggests a more balanced usage of POS types for describing a wider range of phenomena. Notably, the primary and secondary POS categories for KW$_{flood}$ and KW$_{other}$ are diametrically opposed. This can be attributed to KW$_{flood}$'s focus on action and processes related to flooding, while KW$_{other}$ places greater emphasis on specific entities such as "Roads" (道路), "Airplanes" (飞机), and "Buildings" (房屋).

**Table 4.** Distribution of part-of-speech across keyword types.

| Category | Top5_POS | Proportion | Count | Entropy |
|---|---|---|---|---|
| Rainfall | n | 0.69 | 109 | |
| | i | 0.1 | 16 | |
| | v | 0.09 | 14 | 1.48 |
| | l | 0.07 | 11 | |
| | t | 0.04 | 7 | |
| Flood | v | 0.45 | 69 | |
| | n | 0.31 | 47 | |
| | l | 0.07 | 11 | 2.19 |
| | vn | 0.05 | 7 | |
| | i | 0.03 | 5 | |
| Other | n | 0.37 | 104 | |
| | v | 0.26 | 73 | |
| | l | 0.05 | 14 | 2.95 |
| | nr | 0.05 | 14 | |
| | a | 0.04 | 12 | |

Notes: Top5_POS refers to the top five parts of speech (POS), ranked according to the proportion of part-of-speech for all keywords within each category. n: noun; i: idiom; v: verb; l: temporary marker; t: time word; vn: verb noun; nr: proper noun; a: adjective. Entropy is computed by multiplying each part-of-speech type's probability by the logarithm of that probability, summing these values for all types, and then taking the negative. Higher entropy values indicate a more complex and diverse distribution of POS.

### 4.3. Spatial Feature Variations of Keywords

Cities employ varied keywords to characterize rainfall and floods, with the diversity in terminology somewhat illustrating the linguistic richness showcased in Figure 6. Major urban conglomerates such as the Beijing-Tianjin-Hebei region, the Yangtze River Delta, and the Pearl River Delta, as well as noteworthy inland capitals such as Shijiazhuang, Xi'an, Changsha, Guangzhou, and Kunming, display a concentration of keywords depicting perceptions of rainfall. In contrast, areas receiving substantial annual precipitation, exceeding 1600 mm, with a history of flooding disasters, are hubs for diverse flood-related terms. It is noteworthy that other relevant keywords permeate both the major urban landscapes as well as cities known for receiving significant annual rainfall, indicating a broad spatial dissemination of these linguistic elements.

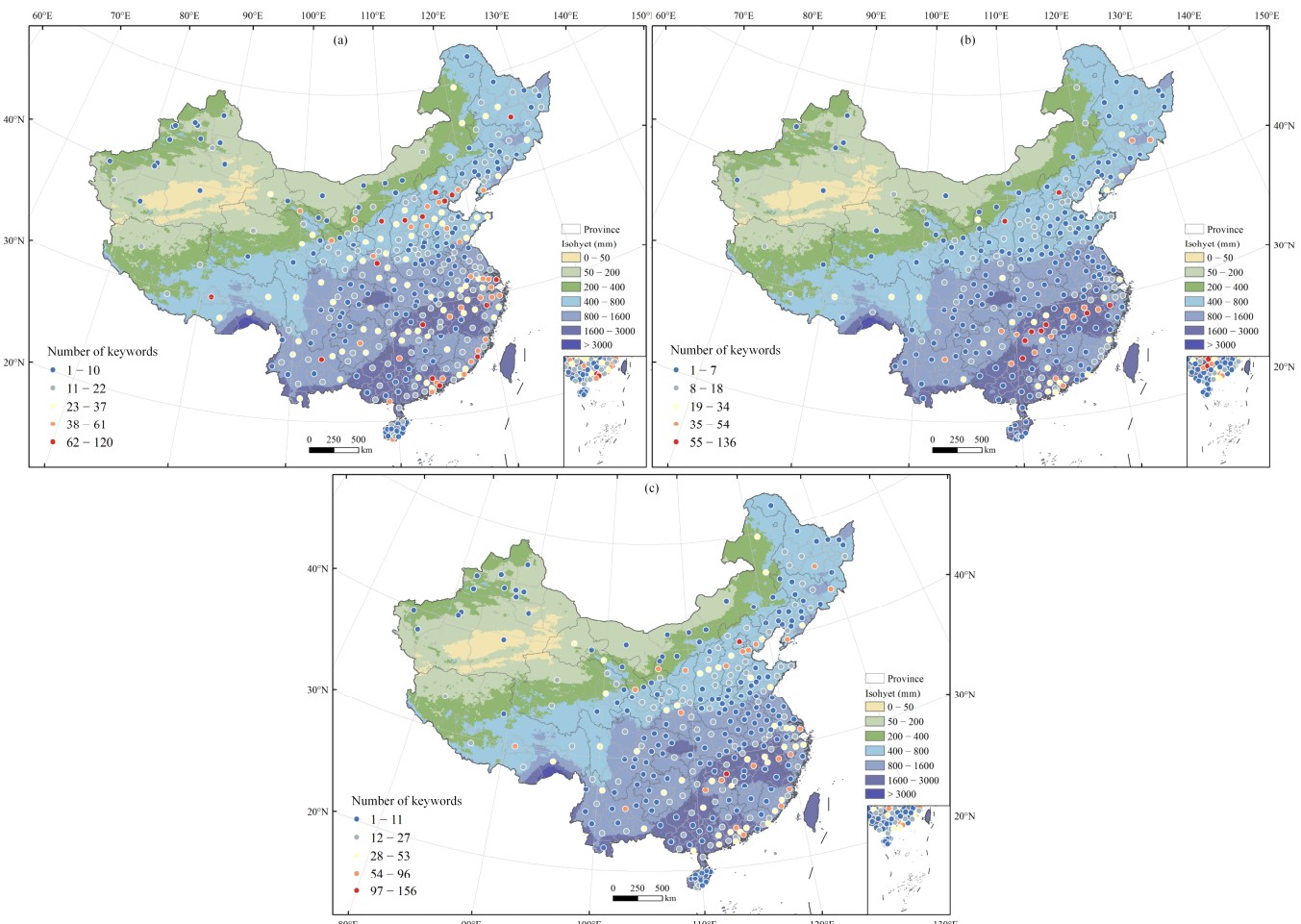

**Figure 6.** The number of keywords per city by type: (**a**) KW$_{rainfall}$, (**b**) KW$_{flood}$, (**c**) KW$_{other}$. Colors in the base map represent the isolines of the average annual rainfall.

Each keyword demonstrates varying significance levels across cities, prompting an investigation into potential spatial clustering patterns among these cities. Initially, we utilized the global Moran's I index to determine the spatial correlation of each keyword among all cities with significant correlations. Of the 594 keywords analyzed, nearly half (49.5%) showed a marked spatial correlation. Among these, KW$_{flood}$ stands out with 60% of its keywords revealing significant spatial correlations, as visualized in Figure 7. This rate is noticeably higher than its counterparts without such correlations. KW$_{rainfall}$ displays an almost equal distribution of significant and non-significant keywords. On the other hand, KW$_{other}$ has the smallest proportion of keywords (43%) with notable spatial correlations, a figure considerably lower than those without such correlations.

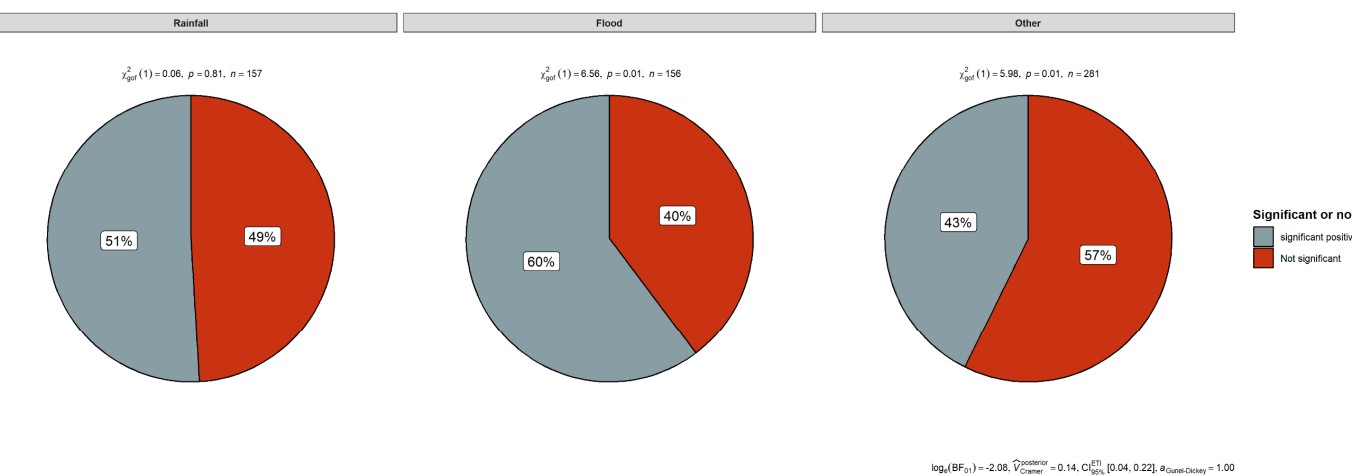

**Figure 7.** The proportion of keywords with significant spatial correlation across keyword types.

Keywords with significant spatial correlation may exhibit regional heterogeneity in their expression (Figure 8). Some keywords, such as Downpour, are commonly used nationwide. However, others are region-specific, where the perception and usage of the keyword varies by region. For instance, "Autumn rain" (秋雨) mainly refers to the precipitation that falls in the northern region of China, while "Plum rain" (梅雨) pertains to the rainy season that predominantly occurs in the middle and lower reaches of the Yangtze River and southern China. On the other hand, Flood typically distributes in areas with an annual rainfall of over 1600 mm. Lastly, "Falling rain" (落雨) and "Soaked" (水浸) are terms unique to the Pearl River Delta region.

To delve deeper into the urban spatial correlation evident in the spatial distribution of keywords, we utilized the median center method, homing in on the central points of cities intimately connected with each keyword topic. $KW_{rainfall}$ central points demonstrated the highest level of clustering, with a peak Moran I of 0.18 and a Z-score of 9.97 (Table 5). The central point for $KW_{rainfall}$ predominantly resides in the Huazhong division plain of China, an area characterized by an annual average rainfall fluctuating between 400 and 1600 mm (Figure 9a). Additionally, the median centers for $KW_{rainfall}$ keywords, which exhibit agglomeration patterns and a larger urban cover (indicated by city number), including terms such as "Rain" and "Rainy", are principally found in Henan and Hubei provinces. Contrastingly, keywords with agglomeration patterns and lesser urban coverage exhibit a more dispersed distribution; for instance, the centers for terms such as "Falling rain" (落雨) and "A shower" (一阵雨) are situated in Guangdong province. In addition, the spatial distribution of $KW_{flood}$ central points is markedly clustered (with a Moran I of 0.12 and a Z-score of 7.78), albeit less so than $KW_{rainfall}$. The focal point for $KW_{flood}$ is predominantly found further south, notably in the mid-to-lower reaches of the Yangtze River, a region experiencing an annual average rainfall exceeding 1600 mm (Figure 9b). The median centers for $KW_{flood}$ keywords, which exhibit agglomeration patterns and a higher urban cover, such as "Flood" and "Flooded", are primarily concentrated in Hunan province. Lastly, the spatial distribution linked with $KW_{other}$ central points also manifests clustering but to a lesser degree, as evidenced by the minimum Moran I value of 0.09 and a Z-score of 5.68. Notably, the spatial distribution of central points linked to $KW_{other}$ keywords mirrors those associated with $KW_{rainfall}$ keywords (Figure 9c), showcasing a similar pattern in their geographical alignment.

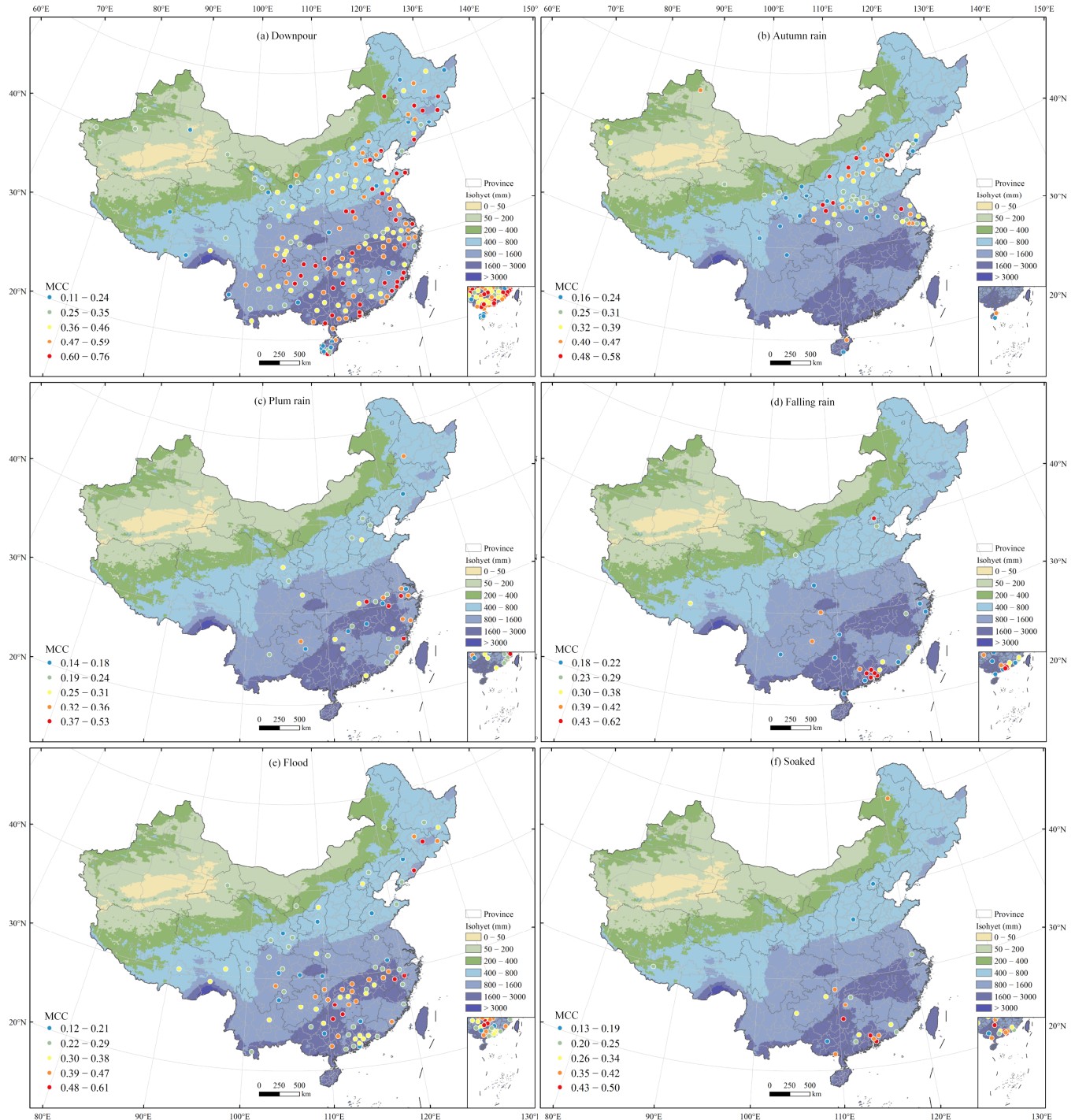

**Figure 8.** The distribution of cities for keywords of significant spatial correlation. Colors in the base map represent the isolines of the average annual rainfall.

**Table 5.** Spatial autocorrelation of median centers of the cities significantly correlated with each keyword across keyword types.

| Category | Moran I | Z-Score |
|---|---|---|
| Rainfall | 0.18 | 9.97 |
| Flood | 0.12 | 7.78 |
| Other | 0.09 | 5.68 |

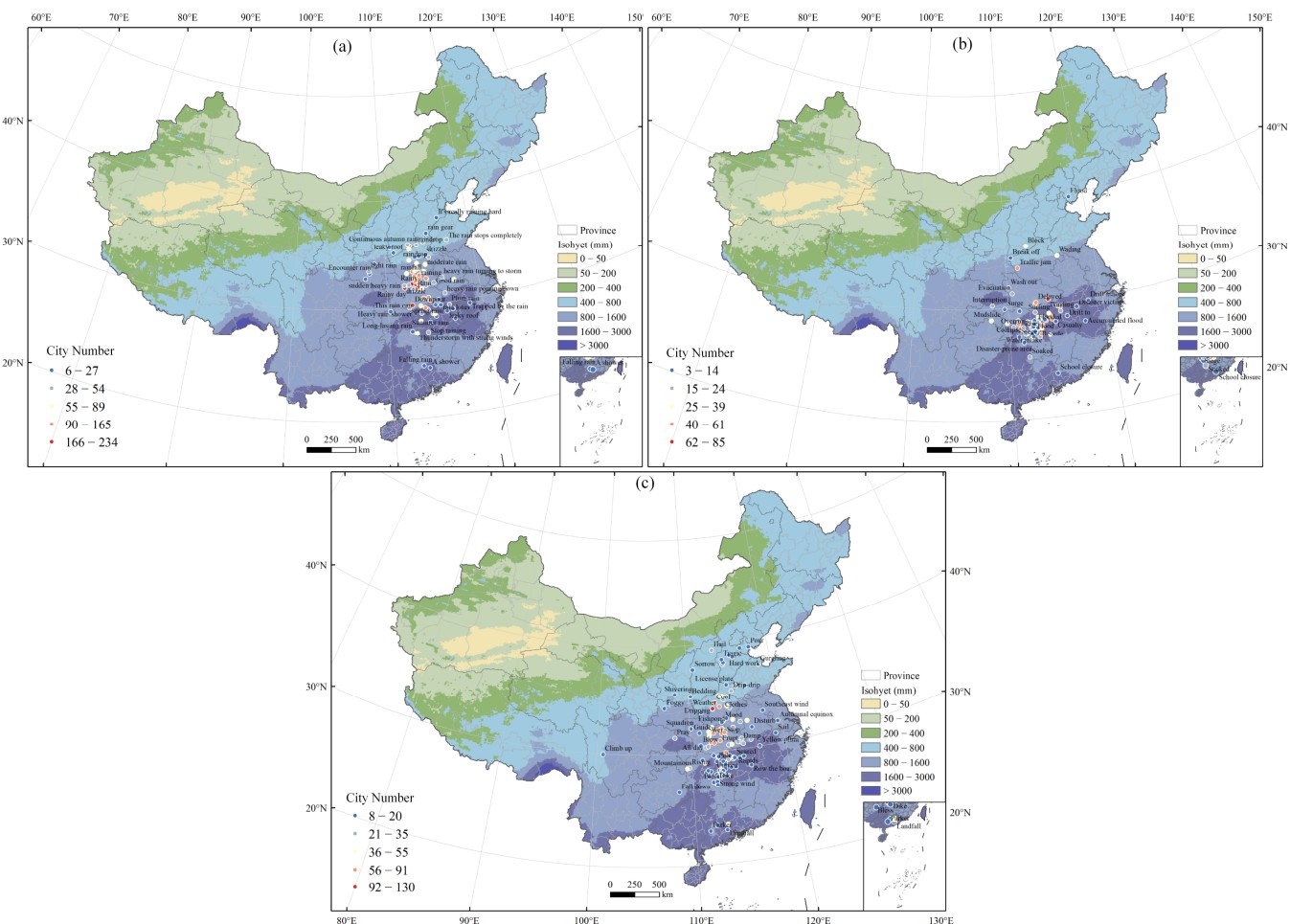

**Figure 9.** The median centers of the cities significantly correlated with each keyword by type: (**a**) $KW_{rainfall}$, (**b**) $KW_{flood}$, (**c**) $KW_{other}$. Colors in the base map represent the isolines of the average annual rainfall. The spatial autocorrelation report is calculated by combining the distribution of central points and corresponding number of cities.

## 5. Discussion

The goal of this study is to measure the diversity of urban linguistic expressions by creating new techniques to capture expressions related to rainfall and flooding. By doing so, we provide a more comprehensive and detailed description of the public's perception and expression when facing rainfall or floods and analyze the regional heterogeneity of this expression. In this section, we will further reveal the mechanisms behind these heterogeneities and variations by discussing the possible factors that may influence the choice of specific public language, as well as the potential factors that enrich urban linguistic expression, and provide some insights and ideas for leveraging language diversity for urban resilience.

### 5.1. Potential Influencing Factors of the Public Choice of Specific Terms

Overall, the potential factors that might influence the public's language choices can be divided into two levels: natural and social (Table 6). The natural level includes two dimensions: rainfall characteristics and weather conditions, while the social level includes education and dialect habits.

**Table 6.** Classification tree of potential influencing factors of keywords.

| Level 1 | Level 2 | Level 3 | Level 4 | Keywords |
|---------|---------|---------|---------|----------|
| Natural level | Rainfall characteristics | Singular rainfall features | Rainfall intensity | Light rain; Moderate rain; Heavy rain; Floods |
| | | Multi-dimensional rainfall | Rainfall duration | Continuous rain; Prolonged rain |
| | | | Rainfall timing | After the rain; Rainy night |
| | Weather conditions | Temperature | | Autumn rain |
| | | Humidity | | Plum rain |
| Social level | | Education level | | Torrential rain pours down |
| | | Dialect habits | | Accumulated water; The water has risen; Soaked; Where did all this water come from? |

Rainfall characteristics are divided into singular rainfall features and multi-dimensional rainfall features. Singular refers mainly to rainfall intensity, different intensities may bring different perceptions and expressions to the public, such as using different terms such as "Light rain", "Moderate rain", "Heavy rain", and "Flood" to describe varying intensities. Multi-dimensional rainfall features include not only rainfall intensity but also rainfall duration, timing, etc. Terms such as "Continuous rain" (连阴雨) and "Prolonged rain" (久雨) are expressions of long-duration, low-intensity rainfall, while phrases such as after the "Rain" (雨后) or "Rainy night" (雨夜) focus more on the timing of the rain.

Temperature, humidity, and other meteorological factors are also important elements influencing the public's language choices under rainy conditions. For example, "Autumn rain" (秋雨) often appears in expressions such as "A cold snap following autumn rain" (一场秋雨一场寒), representing the combined sensation of rainfall and sudden temperature drop. Such temperature changes are mainly found in northern China, so these expressions are also concentrated there. The term "Plum rain" (梅雨) is commonly used by the public during rainy and highly humid weather because it often feels stuffy and muggy. This humidity condition mainly occurs in southern China, so these expressions are also concentrated there.

Education level is also an important factor affecting the public language choices [68]. For example, the phrase "Torrential rain pours down" (暴雨如注) mainly appears in large cities such as Beijing, Shanghai, Guangzhou, and Shenzhen. Residents in these cities have higher cultural education levels, leading to more metaphorical expressions and idioms.

Local dialect habits are also an important factor influencing the public language choices [69]. For example, while "Accumulated water" (积水) is a common term to describe floods, people in Nanchang might say "The water has risen" (涨水), and Cantonese speakers might use "Soaked" (水浸) as in "Heavy rain, waterlogged streets" (落雨大，水浸街). People in the northern dialect areas might say something such as "where did all this water come from?" (哪来这么多水?)

*5.2. Potential Influencing Factors of the Richness of Urban Language Expressions*

Regarding the expression of rainfall, the selection of terminology in large cities is more diverse (Figure 6), possibly due to the following reasons:

Population migration: large cities typically attract more outsiders, who come from various regions and backgrounds, each bringing their unique language and expressions [70]. This cultural blend results in a more varied and enriched language expression in big cities.

Education level: The educational standard in big cities is generally higher. People have received better education, making their understanding and use of language more accurate and precise [71]. This high level of education leads to more enriched language expression in large cities.

Media communication: large cities are often hubs for media, including newspapers, magazines, television, and radio [72]. These media spread information and language ex-

pressions that are more diverse and richer. Exposure to various vocabulary and expressions through media also enhances the richness of language expression in large cities.

Social networks: Social networks in big cities are well-developed, enabling people to easily connect with others from various backgrounds and professions, thus promoting the diversity and richness of language communication and expression [73].

In conclusion, the increased richness in language expression in large cities may result from the combined influence of factors such as population migration, educational standards, media communication, and social networks.

### 5.3. Leveraging Language Diversity for Urban Resilience

Our research introduces a nuanced perspective on urban management by demonstrating how language diversity can sharpen the accuracy and sensitivity of public perceptions regarding rainfall and flood events. This approach fosters a more resilient and sustainable model of urban governance, encapsulated in three key phases: precise rainfall assessment, effective disaster response, and comprehensive post-disaster recovery.

First, leveraging the variety of languages in describing rainfall events enables a more accurate public understanding, enhancing the classification of rainfall intensity beyond simple physical measurements. This nuanced perception is crucial for issuing targeted weather alerts and services that meet the specific needs of diverse demographic groups, thereby improving public preparedness and safety.

Second, recognizing and understanding localized language expressions are vital in crafting effective disaster response strategies, especially in regions with multiple dialects. This ensures that emergency planning is inclusive, preventing the oversight of local needs and promoting more efficient, community-focused responses.

Third, the rich diversity of language expressions in the aftermath of disasters provides insights into the public's emotional states and needs, enabling the development of more grounded and effective recovery strategies. Such strategies support urban resilience by ensuring that recovery efforts are closely aligned with the actual experiences and preferences of affected communities [74].

Incorporating language diversity into urban management and environmental policy-making results in more effective and nuanced strategies across all disaster management stages. This approach marks a significant advancement in urban governance, promising enhanced growth and resilience for cities. By catering to the linguistic needs of diverse communities, we can foster a more informed, prepared, and resilient urban populace, contributing significantly to global efforts against climate change and its impacts.

### 5.4. Limitations and Future Directions

The study, while offering valuable insights, has room for enhancement in its demographic representation. According to the "2020 Weibo User Development Report", a significant portion of Weibo's user base skews younger, with about 80% being from the post-90s and post-00s cohorts [75]. Consequently, the findings primarily reflect linguistic practices among young and middle-aged urbanites, leaving a gap in understanding the language use among other critical demographic groups such as the elderly, children, and residents of rural areas. This demographic skew may limit the generalizability of the study's conclusions across broader populations. Future research should aim to diversify sample populations to include these underrepresented groups, thereby offering a more comprehensive view of linguistic patterns and preferences. Expanding the demographic reach would not only enrich the dataset but also enhance the applicability and relevance of the research findings to a wider audience, providing deeper insights into language evolution and usage across different age groups and living environments.

Moreover, the study overlooks the influence of climatic conditions on the interpretation of weather-related keywords, which can vary significantly across different regions. For instance, the term "Downpour" might be perceived differently by residents of arid areas, who may use it to describe less severe precipitation events, compared to those liv-

ing in humid regions. Research by Qian, Du [36] highlights substantial spatial differences in the perception thresholds of heavy rain across China, with higher thresholds observed in southern cities compared to northern ones, and even surpassing the standard 50 mm heavy rain warning criteria. Coastal cities, especially in the southeast, exhibit lower perception thresholds due to the compounded risks of heavy rains and storm surges, leading to potential disruptive flooding. Future research endeavors should delve into the complex interactions between climatic conditions and the usage of specific keywords, particularly those related to extreme weather phenomena. Such investigations would provide nuanced insights into how language adapts to environmental conditions, enriching the discourse on linguistic practices within the context of climate change.

## 6. Conclusions

The study constructs the keyword library connected to rainfall and floods, and analyzes the semantic characteristics and regional variations of various keywords. By using social media Weibo data in 2017, the study offers the following insights:

Firstly, we present a novel algorithm that identifies linguistic expressions related to rainfall and floods, taking into account the connection between precipitation observations and linguistic expressions. Implementing this algorithm on 210 million social media entries from 2017, we identified 594 keywords related to rainfall and flooding. This count is 20 times higher than the typical manually created bag-of-words used in many studies, significantly enriching the keyword library. These results lay a robust foundation for research that employs social media data to explore public perceptions of rainfall and flood events. Particularly in linguistically diverse regions such as China, our algorithm enhances the comprehensiveness and precision of such studies.

Secondly, we conduct a comprehensive analysis of three types of keywords' semantic attributes, including popularity, credibility, time delay, and part-of-speech. Among the three categories, rainfall-related keywords are the most widely used, notably "Rainy" (下雨), "Rain" (雨), "Downpour" (暴雨); flood-related keywords often have the longest delay, with perceptions of flood damage typically delayed by about a day from the actual peak rainfall. Specifically, for keywords such as "Flooding", "Post-disaster" (灾后), and "Supplies" (物资), the delay can extend beyond 2 days. There are significant differences in the part-of-speech for keywords across different categories; $KW_{rainfall}$ is primarily a noun, whereas both $KW_{flood}$ and $KW_{other}$ consist of verbs and nouns, but with completely opposite primary and secondary roles.

Thirdly, we analyze spatial characteristics from two perspectives: the keyword viewpoint (considering regional applicability and global or local nature) and the city-centric view (evaluating keyword diversity within cities). Among the 594 keywords, 49.5% show significant spatial correlation, reflecting potential regional variations in expression. For instance, "Autumn rain" (秋雨) refers mainly to precipitation in northern China, while "Plum rain" (梅雨) pertains to the rainy season in the Yangtze River's middle and lower reaches and southern China. Major urban centers, such as Beijing-Tianjin-Hebei, Yangtze River Delta, Pearl River Delta, and cities such as Shijiazhuang, Xi'an, Changsha, Guangzhou, and Kunming, show a higher concentration of rainfall-related keywords. Conversely, areas with varied flood-related expressions are mainly found in regions with over 1600 mm of annual rainfall and a historical prevalence of flood disasters.

**Supplementary Materials:** The following are available online at https://www.mdpi.com/article/10.3 390/ijgi13030092/s1, Figure S1: The distribution of the number of microblogs through preprocessing and filtering, Figure S2: Keywords of dominated categories of part-of-speech across keyword types.

**Author Contributions:** Jiale Qian, Yunyan Du, Fuyuan Liang and Jiawei Yi conceived and designed the study and methods; Jiale Qian, Yunyan Du, Nan Wang, Wenna Tu, Sheng Huang, Tao Pei and Ting Ma analyzed the data; Jiale Qian, Fuyuan Liang and Yunyan Du wrote the paper, and all coauthors contributed to the interpretation of the results and to the text. All authors have read and agreed to the published version of the manuscript.

**Funding:** This research was funded by the Key Project of Innovation LREIS (KPI002), National Science Foundation of China (42176205).

**Data Availability Statement:** The keywords library is directly accessible from the journal website. Other data presented in this study are available on request from the corresponding author.

**Conflicts of Interest:** The authors declare no conflicts of interest.

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
