# Peer review of "Quantifying Urban Linguistic Diversity Related to Rainfall and Flood across China with Social Media Data"

_ijgi, doi:10.3390/ijgi13030092_

Round 1

Reviewer 1 Report

Comments and Suggestions for Authors

This work, which has good practical value but still has some issues, quantitatively analyzes the urban linguistic variety of rainfall and flood using data from social media.

The following are the questions:

1.There isn't a line number on the manuscript.

2. It is advised to provide the website, the date of access, the precise source of the data, etc. in Section 2. A tabular listing of the data is advised.

3. What is the novel aspect of this approach as stated in section 3? It is advised to include the technique flow chart, etc., to explain the procedure.

4. The review is impacted by the inaccuracy in the final sentence of Section 3.1.

5. A conclusion is given in both Section 4 and Section 6.

6. Section 4.1: Where do the terms "0.66 and 0.48," "0.48 and 0.46," and "0.37 and 0.35" originate from in the paragraph above Figure 2?

7. All of the figures in the text, with the exception of Figure 3, are too hazy to read, and it is advised that the high-definition figures be replaced.

8. Placing the diagram close to the descriptive text is advised. It is advised to review and edit the entire document.

9. To what do the letters in Table 3's Top5_POS column correspond?

10. Are some of the paragraphs 2 and 3 in Section 4.3 duplicated?

11. The Z-score and Moran I values in section 4.3, paragraph 5 recommend adding data tables.

Reviewer 2 Report

Comments and Suggestions for Authors

Throughout the paper, where possible, please consider explaining technical terms and acronyms (e.g., ArcGIS, GPM, MCC, OL, MBk, WBK, Global Moran's I index, ISPRS, POS, FIRE, Microblogs), which will make the content more accessible to a wider audience.

Tables

There are four tables in the paper. Here are suggestions you should consider:

Table 1: 

  • The presentation of Pearson correlation coefficients and P-values should be consistent in terms of formatting, such as using a uniform number of decimal places.

  • Provide information or a footnote explaining the significance of the Pearson r values and P-values.

Table 2: 

  • Briefly explain in the table's footnote what ‘True’ and ‘False’ mean in this context. In other words, clarify the criteria used for determining the relevance of keywords to precipitation (True/False column).

  • Standardize the presentation format of the Maximum Coefficient of Correlations and Optimal Latency values.

  • Provide a brief explanation or footnote about the algorithm used for filtering keywords.

Table 3: 

  • Provide a brief explanation or footnote about Part-of-Speech Categories. Categories like nouns (N), verbs (V), adjectives (Adj), adverbs (Adv), interjections, conjunctions, etc. should be explained.

  • Provide at least a basic explanation of how entropy values are calculated.

Table 4: 

  • Provide clear and concise explanations for each of the five levels of classification. In particular, if the five-level classification is a novel approach developed for this study, you should certainly explain it.

Figures

There are eight figures in the paper. All figures look crowded due to being packed into a small space. Here are suggestions you should consider:

Figures 1, 2, 4, and 6:

  • Increase the size for better visibility and readability of details.

  • Ensure that the graph legends and labels are clear and easily understandable.

  • Consider adding a brief explanatory note or caption for clarity.

Figure 3:

  • Ensure that the word cloud is large enough to read all words clearly.

  • A caption explaining the methodology and relevance of the word cloud would be helpful.

Figures 5, 7, and 8:

  • Crowding too many maps in a small space can make the figure look cluttered and overwhelming. Ensure the map is of adequate size and resolution for detail visibility.

  • Labels and legends should be legible and informative.

2. Data

Page 3: The collection of over 210.8 million geotagged microblogs from 2017 is impressive and provides a rich dataset for analysis. Yet, please consider elaborating on how it ensured the representativeness of this data across different regions and demographic groups. 

In addition, the dataset includes various metrics like user-reported locations, timestamps, and retweet metrics. You may consider a more detailed explanation of how each of these metrics was used in the analysis so that it may help the reader to understand the study's methodology. 

Finally, The study touches upon data collection but lacks a clear explanation of how it handled this extensive dataset or addressed privacy concerns related to user data from social media. I recommend including a more detailed description of these aspects for greater clarity.

3. Method: 

Pages 4 to 5: The complex methodology might be difficult for readers outside the specialized field to grasp. To resolve the issue, I suggest you consider incorporating diagrams, flowcharts, or infographics that visually represent the algorithm's workflow. 

4. Conclusion:

On page 5, I recommend renaming “4. Conclusion” to “4. Results” to avoid confusion, as there is already a separate conclusion section labeled as “6. Conclusion.”

The introduction of metrics like the Maximum Coefficient of Correlations (MCC) and Optimal Latency (OL) is valuable. However, you may consider providing a concise explanation of how these metrics help in identifying precipitation-linked keywords.

While the section provides technical details of data processing and filtering, it appears that it does not strongly connect these steps with the broader research goals and objectives. I suggest you consider describing how these filtering techniques contribute to understanding linguistic diversity in the context of rainfall and floods.

5. Discussion: 

Within pages 12 to 14, since the study relies on social media data, it may not fully represent the population's views. I believe that discussing these limitations and their implications could strengthen the study. It also needs to provide more insight into how this research can be applied in environmental policy-making or in strategies for increasing public awareness.

6. Conclusion:

On page 15, the conclusion predominantly reiterates the findings without offering a reflective analysis or discussing the implications of these findings in a broader context. You may consider including insights into how these findings could impact future research or policy-making. 

While the study's achievements are highlighted, there is a lack of discussion on its limitations and potential areas for future research. You may consider acknowledging the study’s limitations and suggesting future research avenues. 

The conclusion mentions the novel algorithm and its implementation. It may need a critical reflection on the methodology used, including its strengths and potential areas for improvement.

Reviewer 3 Report

Comments and Suggestions for Authors

The article introduces an algorithm that captures expressions related to rainfall and floods within a large volume of social media entries. Identified keywords and lexical expressions are further analyzed in terms of their occurrence frequency in microblogs, in terms of their grammatical category, their credibility and time delay. Finally, some spatial characteristics of the keywords are assessed showing the regional variations across language usage.

The main idea behind the proposed approach is to illustrate the language diversity across China for issues related to floods and rainfalls. Such demonstrations besides verifying the linguistic variations exemplifies the driving forces behind these variations and it could serve as a guide towards providing regional equalities in environmental / disaster management.

The article addressed sufficiently related works and advances existing research by exploring linguistic diversity in a new domain as well as by employing a big dataset for applying the proposed algorithm.

However, the proposed rule-based algorithm for identifying keywords (p.4) is not adequately described. the p-values upon which the algorithm operates are not justified. Moreover, it is not explained whether an existing algorithm could be adapted to this specific domain so that there is less effort associated with coming up with a new tool.

Nevertheless, the algorithm's outcomes are thoroughly discussed throughout the article and several details and examples are given with respect to the obtained results (e.g. popular keywords, regional variations in language usage, etc.).

An interesting aspect of the article is that the authors justify the lexical variation across China's regions by referring to educations, environmental or cultural issues. Still this justification merits further elaboration and it seems somewhat intuitive,

A weakness of the proposed approach is that it is specifically tailored to a specific domain (floods, rainfall) within a particular geographic region (China). This, coupled with the fact that there is no comparative evaluation of the proposed algorithm restrain the generalization of the proposed approach across different domains.

Reviewer 4 Report

Comments and Suggestions for Authors

The increased emphasis on language and culture preservation has expanded research on linguistic diversity, especially within diverse linguistic and cultural terrain in China

Rainstorms and floods serve as an ideal context for linguistic diversity studies due to their significant financial impacts, especially in countries like China, which vastness, spanning over 9.6 million km2,  encompasses over 80 dialects. Therefore, neglecting linguistic diversity in research or disaster management could lead to regional biases and cause ine-qualities.

The peer-reviewed article sets out to provide a comprehensive quantification of rainfall and flood-related keywords. This research offers three major contributions to the discipline. 

First, the authors introduce a novel algorithm that captures expressions related to rainfall and floods by considering the nexus between precipitation observations and linguistic phrases. Implementing this approach on 210 million social media entries from 2017, the authors identified 594 keywords associated with rainfall and flooding.  This count is 20 times higher than the typical keyword set used in many studies, significantly enriching the keyword library.

Second, the authors conducted a thorough analysis of these keywords' semantic attributes, exploring aspects like popularity, credibility, and time delay. 

Third, the authors dissect spatial characteristics from dual viewpoints: the keyword perspective (considering each keyword's regional applicability and its global or local nature) and the city-centric perspective (assessing keyword diversity within individual cities). Furthermore, the authors discuss the driving forces behind these distributions, enabling  authors to offer insights that can guide local authorities and residents in formulating nuanced strategies and fostering a sustainable habitat.

The results of the article are important because incorporating language diversity into urban management leads to more effective and nuanced strategies across the stages of disaster management. This represents a more intelligent and sustainable approach to urban governance, beneficial for the long-term growth and resilience of cities.

However, the authors did not consider the issue of the dependence of the use of keywords on the climate of the area. It is a plausible hypothesis that, for example, the word “Downpour” is used by residents of arid areas to denote a smaller volume of precipitation than residents of humid areas. The same applies to other keywords denoting abnormal weather phenomena. This hypothesis needs to be tested.

In addition, the article is poorly formatted. The section (number 4 and number 6) is called “Conclusion”. On page 4 there is no bibliographic reference in the sentence "The distribution of microblog counts across cities is illus-trated in..."

The article can be accepted after revision.

Reviewer 5 Report

Comments and Suggestions for Authors

The proposed research is interesting, but the manuscript needs to be organized and structured in a more complete way. First of all, it is necessary in the introduction to better highlight what the critical points of current related research are and how the proposed method addresses them. For example, the research makes use of semantic analysis processes by analyzing texts connected to rainfall and flood-related keywords extracted from social media, but there is a vast literature on semantic analysis methods and many NLP algorithms are used for the analysis of texts and extracting information from social media. What, then, is the added value of research in this area? The authors must clarify these points well.

There is no architectural scheme or flow diagram to add in section 3, showing the composition of the proposed method. The processes described in paragraphs 3.1 - 3.4 should be represented as functional components in this diagram.

At the end of paragraph 3.1 an error appears regarding a reference in the bibliography not found. It is necessary to correct this anomaly.

For greater clarity, it is useful to add the proposed algorithm in pseudocode in order to better outline the parameters, the functions used, the input information, the logic of the method and the output information generated. I recommend adding it to the end of section 3.

How is the entropy of individual categories calculated? Authors must specify which concept of entropy they are referring to and how they calculate this measure.

What are the future prospects for research? It is necessary to add a discussion on future prospects in the final section.

Comments on the Quality of English Language

Some grammar typos appear in the text. They must be removed.

Round 2

Reviewer 1 Report

Comments and Suggestions for Authors

Based on previous feedback, it seems like the authors made improvements and corrections to the manuscript in addition to adding some valid justifications and additions.

1.Furthermore, "expression" should be used instead of "expr-ession" in Figure 1.

2.The amount of wording duplication in the manuscript is 28%, it must be reduced.

Reviewer 2 Report

Comments and Suggestions for Authors

I think that the revisions have enhanced the paper's clarity and coherence. Additionally, the incorporation of an additional flowchart and supporting sections such as  “Limitations and future directions” has strengthened the paper's persuasiveness and depth of analysis. Overall, these revisions have changed the paper into a more impactful piece of writing.

Reviewer 3 Report

Comments and Suggestions for Authors

Thank you for addressing the review comments in the revised version of the article. Now, the content has been improved and with the various additions and clarifications given its added value has increased.

Reviewer 4 Report

Comments and Suggestions for Authors

The authors added a new section, 5.4 "Limitations and Future Directions," where they delved into the issue of the importance of considering the influence of regional climates on the usage of keywords related to rainfall and floods. Thus my comments were taken into account by the authors.

The article can be accepted in the pesent form.
